# Chronic Inflammatory Demyelinating Polyneuropathy after ChAdOx1 nCoV-19 Vaccination

**DOI:** 10.3390/vaccines9121502

**Published:** 2021-12-19

**Authors:** Caterina Francesca Bagella, Davide G. Corda, Pietro Zara, Antonio Emanuele Elia, Elisa Ruiu, Elia Sechi, Paolo Solla

**Affiliations:** 1Azienda Ospedaliero Universitaria Sassari, Unit of Neurology, Department of Medical, Surgical and Experimental Sciences, University of Sassari, 07100 Sassari, Italy; pietrozara93@gmail.com (P.Z.); elisa.ruiu@tiscali.it (E.R.); eliasechi87@gmail.com (E.S.); paolo.solla@aousassari.it (P.S.); 2Parkinson and Movement Disorders Unit, Department of Clinical Neurosciences, Fondazione IRCCS Istituto Neurologico Carlo Besta, 20133 Milan, Italy; antonio.elia@istituto-besta.it

**Keywords:** chronic inflammatory demyelinating polyneuropathy, Guillain–Barrè syndrome, bifacial weakness, COVID-19, SARS-CoV-2 vaccine

## Abstract

Recently several patients, who developed Guillain–Barré syndrome characterized by prominent bifacial weakness after ChAdOx1 nCoV-19 vaccination, were described from different centers. We recently observed a patient who developed a similar syndrome, later in the follow up he showed worsening of the neuropathy two months after the initial presentation. Repeat EMG showed reduced nerve sensory and motor conduction velocities of both upper and lower limbs, and a diagnosis of chronic inflammatory demyelinating polyneuropathy (typical CIDP) was made according to established criteria. Our report expands on the possible outcomes in patients who develop Guillain–Barrè syndrome after COVID-19 vaccinations and suggest that close monitoring after the acute phase is needed in these patients to exclude a chronic evolution of the disease, which has important implications for long-term treatment.

## 1. Introduction

Vaccines are the most cost-effective measures available against the COVID-19 pandemic with more than seven billion doses administered worldwide in November 2021 (https://coronavirus.jhu.edu/map.html) (accessed on 11 December 2021), thus monitoring of rare adverse events is of fundamental importance. COVID-19 vaccinations have mostly been associated with adverse events of little clinical significance such as injection site reaction or non-specific systemic symptoms such as fever and muscle aches, which usually resolve within days [1]. However, more severe adverse events have been reported in a minority of cases, including immune-mediated thrombocytopenia [2], and Guillain–Barrè syndrome, that is a potentially serious complication with a mortality rate of 3–5% despite intensive care treatment [3].

The incidence of GBS is estimated to be around 0.4–4.0 cases per 100,000 person-years [4]. The disease is more common in males especially over the age of 75, and less common among children [5]. GBS is generally classified into various subforms based on the pathophysiological profile and symptomatology (e.g., pharyngeal-cervical weakness syndrome, brachialis and bifacial paralysis with paresthesia) [6]. In some patients GBS can deteriorate again after some weeks from onset and several relapses can occur; in these patients a diagnosis of acute-onset chronic inflammatory demyelinating polyneuropathy (CIDP) should be considered [7].

In GBS an infectious trigger is recognized in >50% of cases while onset following vaccinations is debated [4]. Recently several patients who developed a GBS following CoV-19 vaccinations, were described from different centers [8,9,10,11,12]. In particular, most patients developed bifacial weakness with or without involvement of other peripheral nerves after ChAdOx1 nCoV-19 vaccination, suggesting this might represent a specific neuropathy phenotype associated with the vaccine. Clinical features of these patients are summarized in Table 1.

We recently observed a patient who developed a similar acute syndrome after the first dose of the ChAdOx1 nCoV-19 vaccine, but with worsening of the neuropathy two months after the initial presentation leading to a final diagnosis of CIDP.

## 2. Case Presentation

The patient was a 49-year-old man who presented asymmetric bilateral facial weakness, and paresthesias in the tongue and face. Sixteen days before symptoms onset he received the first dose of ChAdOx1 nCoV-19 vaccine. He denied infections within the prior month. Neurologic examination showed severe bilateral facial paresis, more prominent on the right side, and lower limbs areflexia. Brain and spinal cord MRI showed enhancement of the facial nerves and of the cauda equina and lower thoracic nerve roots, respectively (Figure 1). Cerebrospinal fluid analysis revealed elevated proteins (110 mg/dL; normal range <45 mg/dL) without pleocytosis, and absence of intrathecal IgG synthesis. Serum search for ganglioside autoantibodies was negative, including anti-GQ1b, anti-GD1b, and anti-GM1 antibodies. Blink reflex test displayed bilateral impairment of the motor efferent pathway, while the remaining nerve conduction studies (NCS) and electromyography (EMG) examination were normal. A microbiological work-up of cerebrospinal fluid has ruled out infection of campylobacter jejuni, Epstein–Barr virus, cytomegalovirus, influenza A virus, and hepatitis A, B, C virus. An initial clinical diagnosis of GBS was made and intravenous immunoglobulins (IVI g; 0.4 g/kg/die for 5 days) were initiated 3 days after symptoms onset with mild improvement of the bifacial paresis and resolution of the hypoesthesia.

Two months after presentation, his symptoms worsened with painful paresthesias and progressive lower limbs weakness, more prominent during right foot dorsiflexion (4/5 MRC). He developed mild sensory ataxia with wide based gait and postural instability. Repeated nerve conduction study showed marked slowing of sensory and motor conduction velocities of both upper and lower limbs, prolonged distal latency, conduction blocks and absent F response, leading to a diagnosis of chronic inflammatory demyelinating polyneuropathy (CIDP) according to established criteria [13]. He received a second cycle of IVIg with improvement of paresthesias.

Later he was treated with IVIg cycles every six weeks. At last clinical evaluation, six months after presentation, the clinical features were characterized by complete resolution of left facial palsy with persistence of slight weakness in the right upper facial area. Gait ataxia improved but did not resolve completely. Paresthesias on the lower limbs were disabling and thus treatment with pregabalin and alpha lipoic acid was started with benefit. Clinical features and examinations of the patient are summarized in Table 1.

## 3. Discussion

Acute demyelinating polyneuropathy is a rare, but well recognized complication of COVID-19 vaccines. The disease variant with bifacial weakness has recently been recognized to be more common in patients with adenoviral-vectored (Oxford AstraZeneca, Johnson and Johnson) vs. mRNA-based (Pfizer BioNTech, Moderna) vaccines [14]. In line with these findings, our patients developed an acute demyelinating polyneuropathy with bifacial palsy 2 weeks after the ChAdOx1 nCoV-19 vaccine, but subsequent clinical and EMG worsening prompted a diagnostic revision to CIDP with acute onset. To date, only one patient who developed CIDP after COVID-19 vaccination was reported, with typical gradual onset of ascending lower limb weakness and sensory changes without facial involvement [15].

A difference of the previously described cases of bifacial palsy, in which the clinical course is reported favorable or stabilized or which have been described in the acute phase, in our patient is available the most prolonged follow-up, that is of 6 months. This pro-longed clinical observation led to diagnose a chronic form of demyelinating polyneuropathy, which was an unexpected clinical evolution. Therefore, our report expands on the possible outcomes in patients who develop Guillain–Barrè syndrome after COVID-19 vaccinations and suggest that close monitoring after the acute phase is needed in these patients to exclude a chronic evolution of the disease, which has important implications for long-term treatment.

Our patient presented a clinical onset of GBS with bifacial weakness with paresthesias phenotype; 8 weeks later he presented a clinical deterioration, thus a diagnosis of CIDP was made, supported by neurophysiological findings. The distinction between GBS with fluctuations after start of treatment and CIDP is difficult but it has been recommended that diagnosis of CIDP should be considered when a patient thought to have GBS deteriorates again after 8 weeks from onset or when deterioration occurs 3 times or more [7].

In GBS, acute motor neuropathy may involve facial nerves in various clinical presentations that include unilateral, bilateral facial palsy, bifacial palsy associated with involvement of other cranial nerves and bifacial palsy associated with limb paresthesia, moreover facial involvement is also possible in patients with classical forms of GBS. Interestingly it has been proposed that Bell’s palsy may share similar pathogenic mechanisms with autoimmune neuropathies [16].

The specific clinical phenotype of bifacial weakness with paresthesias is quite uncommon and accounts for about less of 5% of the patients affected with GBS or Miller Fisher spectrum disorders [17], whereas in those patients who have a fluctuating GBS with treatment related fluctuations, who later may become a CIDP, facial involvement was reported to be very common and it is present in about 63% of the patients [7]. On the other hand, CIDP at onset presents uncommonly with cranial nerve involvement, which is reported in about 11% of typical forms mostly with ophthalmological implications rather than facial palsy [13].

GBS and CIDP are immune-mediated disorders in which the exact mechanism of the immune response linked to the pathogenesis is still not clear. In GBS, the pathogenic events are associated with the immune response usually to a previous infection, which subsequently leads to a cross-reaction with the production of antibodies directed to epitopes of the myelin sheath or to peripheral nerves and roots, thus causing an autoimmune process. Antecedent infections are reported in about two-thirds of patients with GBS [17], therefore molecular mimicry between microbial proteins and nerve cell surface has been one of the suggested pathophysiologic mechanisms. CIDP is typically an idiopathic disease, in which an immune response targeting myelin components of the peripheral nervous system develops due to an autoimmune process, which mainly involves immune mechanisms mediated by T cells. In CIDP, the finding of an infection that precedes the onset is rare and unlikely [18]. Numerous antibodies are associated with GBS (such as GD1a, GD1b, GM1, GQ1b), while usually no autoantibodies are identified in CIDP patients, even if variants with antibodies directed against myelin or proteins localized to the Ranvier node have been also described (such as neurofascin, gliomedin, and contactin-1), which appear to result in a more severe disease phenotype [18].

Neuropathies after vaccination are rare event and CIDP developing in the postvaccination period is distinctly unusual, accounting for about 1.5% of patients [19], and poorly described [20]. In our patient, the temporal association between the first dose of ChAdOx1 nCoV-19 vaccine and onset of neuropathy is remarkable so that a triggering role of the vaccine appears to be in causal association. It has been proposed that a postvaccination neurological syndrome could result from the generation of host antibodies that cross-react with proteins present in peripheral myelin [8]. These antibodies may be generated in direct response to the SARS-CoV-2 spike protein, because there is evidence of a cross-reactivity between the SARS-CoV-2 spike protein and peripheral nerve glycolipids [8,21]. This hypothesis is supported by reports of some patients with GBS with bifacial weakness and paresthesias subtype also among patients affected by GBS post COVID-19 [22], even if in these series it does not appear to be the most common phenotype [23]. Therefore, an immune response to other components of the vaccine should be considered, such as a response against the chimpanzee adenovirus vector.

The optimal management and long-term outcomes of these patients remain to be determined. We treated our patient according to the available recommendation with IVIg administration at earliest convenience [17], however, he developed treatment related fluctuations as observed in about 6–10% of all patients with GBS [17]. This may be due to the possibility that the treatment effect has worn off while the inflammatory phase of the disease is still ongoing, and thus further treatment with repeating IVIg cycles was administered and nonetheless, the patient developed a form of CIDP. To our knowledge this is the first case, among GBS with bifacial palsy onset post COVID-19 vaccination, that shows this evolution of the disease, on this regard, it would be interesting to have additional details on the clinical and EMG follow-up of the previous reported cases with short follow up [8,9,12].

In recent months, several case reports have shown that some variants of GBS can occur after the administration of the ChAdOx1 nCoV-19 vaccine, and in particular the bifacial weakness and paresthesia appears to be the prevalent form. This case report clearly shows the need for close monitoring of patients developing GBS after COVID-19 vaccination after the acute phase. We highlight that adequate long-term clinical follow-up is essential in these cases, as evolution to a chronic form is possible, despite adequate treatment. While an eventual causal relationship between different types of COVID-19 vaccines and autoimmune neuropathies remains to be determined, neurologists should be aware of this possibility as prompt diagnosis and treatment is crucial to improve outcomes in patients with GBS or CIDP and other potentially serious neurologic disorders.

## Figures and Tables

**Figure 1 vaccines-09-01502-f001:**
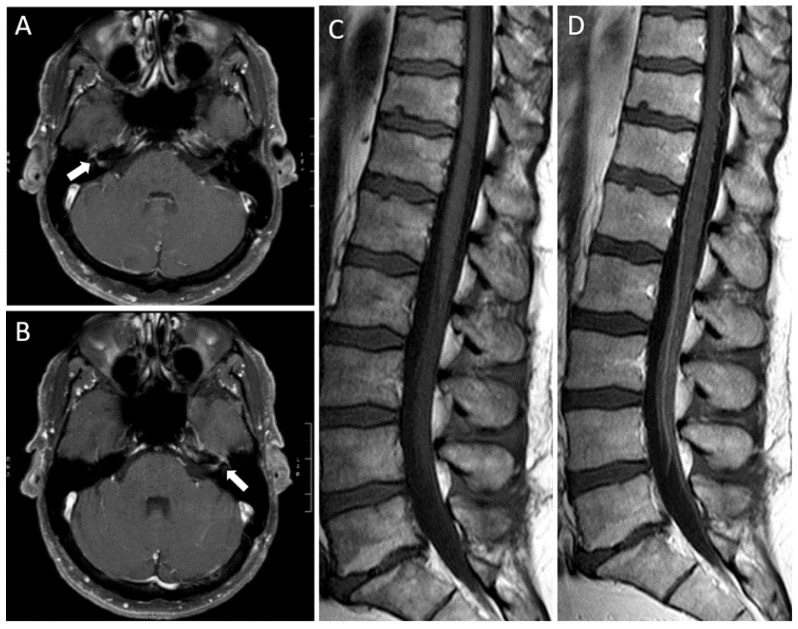
Brain and spinal cord MRI findings. Axial post-gadolinium T1-weighted images of the brain showing enhancement of the right ((**A**), arrow) and left ((**B**), arrow) facial nerves. Sagittal T1-weighted images before (**C**) and after (**D**) gadolinium administration showing diffuse enhancement of the cauda equina and lower thoracic nerve roots.

**Table 1 vaccines-09-01502-t001:** Clinical and MRI characteristics of patients with bifacial weakness after ChAdOx1 nCoV-19 vaccination.

Age/Sex (Authors)	Symptoms/Signs	Respiratory Failure	Days from First Dose of Vaccine	CSF Findings	EMG Findings	MRI	Treatment	Outcome
54/M [8]	Ascending distal limbs dysesthesias; bifacial paresis	None	16	P: 163 mg/dL; C: 19/mL	Day 16: Facial NCS showed severely reduced compound muscle action potential amplitude responses and normal terminal latencies bilaterally; sensory and motor NCS: normal	Enhancement of facial nerves	Prednisolone 60 mg/day × 5 days	Stabilized
20/M [8]	Headache, LL dysesthesias, and bifacial paresis	None	26	P: 123 mg/dL; C: 14/mL	Day 13: Facial NCS showed borderline normal amplitude responses and normal terminal latencies bilaterally; sensory and motor NCS: normal	Normal non-contrast brain MRI	Prednisolone 60 mg/day × 5 days	Stabilized
57/M [8]	Lumbar back pain, dysarthria and bifacial paresis; lower limb dysesthesias; proximal limb weakness on exam	None	21	P: 247 mg/dL; C: 8/mL	Day 13: Facial NCS, not performed; sensory and motor NCS: normal	Normal non-contrast brain MRI	IVIg	Stabilized
55/M [8]	LL paresthesias; bifacial paresis	None	29	P: 89 mg/dL; C: <5/mL	NA	Enhancement of facial nerves	None	Subjective improvement of numbness
43/F [9]	Bifacial paresis, areflexic quadriparesis, upper back pain	Yes	10	P: 72.2 mg/dL C: 5/mL	Demyelinating neuropathy	NA	IVIg IMV	Recovered
67/F [9]	Bifacial paresis, right abducense palsy, bulbar palsy, Distal sensory impairment in the legs, areflexia, limb weakness	Yes	14	P: 345 mg/dL C: 3/mL	Axonal motor-sensory neuropathy	Normal brain and spine MRI	IVIg IMV PLEX	Still hospitalized
53/F [9]	Bilateral LL numbness, weakness, right-sided facial, tongue numbness, and back pain, right trigeminal V2-V3 sensory impairment, areflexia	Yes	12	P: 120 mg/dL C: 3/mL	Demyelinating neuropathy	Normal brain and spine MRI	IVIg IMV	Still hospitalized
68/F [9]	Facial diplegia, bulbar palsy, bilateral facial numbness, bilateral distal lower and UL numbness, and bilaterally trigeminal sensory loss, areflexia	Yes	14	P: 75 mg/dL C: 4/mL	Demyelinating neuropathy	Normal brain and spine MRI	IVIg IMV	Still hospitalized
70/M [9]	Facial diplegia, bulbar palsy. Bilateral distal UL and LL numbness, areflexia	Yes	11	P: NA C: NA	Demyelinating neuropathy	NA	IVIg IMV	Still hospitalized
69/F [9]	Facial diplegia, bulbar palsy, complete ophthalmoplegia, UL and LL distal numbness, UL and LL weakness, areflexia	Yes	12	P: NA C: NA	Demyelinating neuropathy	NA	IVIg PLEX	Still hospitalized
69/F [9]	Facial diplegia, bulbar palsy, bilateral UL and LL numbness, areflexia	None	13	P: 83 mg/dL C: 2/mL	Demyelinating neuropathy	NA	IVIg IMV	Still hospitalized
66/M [10]	Bilateral facial weakness with numbness of the tongue and mouth, interscapular back and LL pain, paresthesia of both hands and feet Normal tone, power and reflexes in both UL and LL, except absent right ankle jerk. Reduced light touch and pinprick sensation symmetrically in LL to the knee and vibration to the ankles. Gait was ataxic	None	7	P: 1.99 g/L C: 2/mL	Sensory NCS: UL and LL: reduced SNAP amplitude Motor NCS: UL and LL: Prolonged DMLs, and F-wave latencies Slow CV Dispersed CMAPs and CB. Facial NCS: Prolonged DMLs	MRI pre and post GAD contrast: normal except for bilateral smooth contrast enhancement along whole facial nerve	IVIg	Facial weakness resolved Pain and paresthesia improving. Intact reflexes including right ankle jerk
43/M [10]	Severe bilateral facial weakness, Myalgia, paresthesia of both hands and feet, severe neck pain, urinary retention, dysphagia, altered taste and paresthesia of tongue Normal limb tone, with full power except mild weakness in right hip flexion. Reflexes initially present but then subsequently lost. Flexor plantar responses. Patchy, asymmetrical glove and stocking reduction in pinprick sensation and a sensory ataxia	None	11	P: 2.81 g/L C: 23/mL	Sensory NCS: UL: absent SNAPs LL: normal Motor NCS: UL and LL: Prolonged DMLs, and F-wave latencies Slow CV Dispersed CMAPs and CB Facial NCS: Absent. Facial EMG: Few fibrillations, no volitional motor units	MRI pre and post GAD contrast: normal except for bilateral smooth contrast enhancement along whole facial nerve	IVIg	20% improvement in facial weakness. Ataxic gait and pain static. Areflexia persists. No longer in urinary retention
51/M [10]	3-week history of severe LL cramping pain. Numbness in feet and hands, spreading proximally to the ankles. Progressive right facial weakness became severe and bilateral after 5 days. Tone, power, and reflexes in limbs were normal. Impaired sensation in all modalities in UL and LL with a sensory ataxia	None	7	P: 5.14 g/L C: 1/mL	Sensory NCS: UL: reduced SNAP amplitudes LL: normal Motor NCS: UL and LL: Dispersed CMAPs Tibial F wave latencies prolonged Facial NCS: Normal except blink reflexes absent Facial EMG: Very reduced volitional motor units	MRI pre and post GAD contrast: normal except for bilateral smooth contrast enhancement along whole facial nerve	None	95% improvement in facial weakness. Ataxic gait 80% better. 25% improvement in pain and paresthesia
71/F [10]	Lower back and abdominal pain. Altered taste and sequential facial weakness within 24 h. Mild proximal LL weakness. Slight weakness in hip flexion bilaterally. Absent knee and left ankle reflexes with normal sensory examination	None	12	P: 0.96 g/L C: 1/mL	Sensory NCS: UL and LL: reduced/absent SNAP amplitudes and velocities Motor NCS: UL and LL: Prolonged DMLs Dispersed CMAPs Facial NCS: Not tested Facial EMG: Not tested	Normal MRI. NO post contrast study. Normal CT performed	None	Residual mild facial weakness, proximal leg weakness and mild paresthesia. Reflexes regained
53/M [10]	Lower back discomfort and radicular pain. Facial, perioral and LL paresthesia progressing to severe simultaneous bilateral facial weakness. Depressed UL reflexes. Normal LL reflexes. Mild distal LL sensory loss to vibration and pinprick	None	8	P: 1.22 g/L C: 0/mL	Sensory NCS: Not tested Motor NCS: Not tested Facial NCS: Not tested Facial EMG: Not tested	Normal MRI. NO post contrast study. Normal CT performed	None	95% resolution of facial weakness, pain and paresthesia
48/M [11]	Severe back pain. Bilateral facial weakness	None	10	P: (1264 mg/L C: 8 × 10 × 6 /L lymphocytes	Severe, multifocal sensorimotor demyelinating polyneuropathy, with reduced compound motor action potentials, reflecting likely hypoexcitability	Normal CT and MRI of the brain	IVIgs Oral Prednisolone	Rapid improvement following the treatment
59/M [12]	Four limb distal paresthesia and postural instability. Bilateral facial palsy (House–Brackmann grade V). Gait ataxia, global areflexia, and distal paresthesia both at the LL and UL; Normal pallesthesia. Segmental strength diffusely preserved (MRC: 5/5). No spine sensory level. No vegetative, or sphincter involvement	None	10	P: 140 mg/dL C: normal white blood cell count	Motor polyradiculoneuropathy with temporal dispersion of the tibial nerve cMAP bilaterally, with F reflex absent in all districts. No sensory involvement, particularly no temporal dispersion of the sural nerve SNAP bilaterally	Unremarkable brain and cervical MRI with gadolinium	IVIg	Slowly improved
49/M (Present case report)	Headache, bifacial paresis and paresthesias; lower limbs areflexia, lumbar back pain	None	16	P: 110 mg/dL C: <5/mL	First admission: Blink reflex: absence of all potentials (R1i, R2i, R2c) with right-sided stimulation and normal findings after left supraorbital stimulation. NCS: absence of demyelinating/axonal neuropathy at upper and lower extremities Second admission: Blink reflex: delay of R1i, R2i after stimulation of left side and R2c delay with right supraorbital stimulation. Absence of R1i and R2i after right stimulation and absence of R2c with left-sided stimulation. NCS: demyelinating sensorimotor polyneuropathy at upper and lower extremities	Enhancement of facial nerves and cauda equina	IVIg	Progressed to CIDP

Abbreviations: C: white cell count; CB: conduction block; CIDP: chronic inflammatory demyelinating polyneuropathy; CMAP: compound muscle action potential; CSF: cerebrospinal fluid; CV: conduction velocities; DML: distal motor latency; EMG: electromyography; F: female; GAD: gadolinium; IVIg; intravenous immunoglobulins; LL: lower limb; M: male; NCS: nerve conduction studies; NA: not available; P: protein levels; PLEX: plasmapheresis; R1i: ipsilateral R1; R2i: ipsilateral R2; R2c: contralateral R2; SNAP: sensory nerve action potential; UL: upper limbs.

## Data Availability

The datasets used and analyzed during the current study are available from the corresponding author on reasonable request.

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
