# Peer review of "Chronic Inflammatory Demyelinating Polyneuropathy after ChAdOx1 nCoV-19 Vaccination"

_vaccines, 2021, doi:10.3390/vaccines9121502_

Round 1

Reviewer 1 Report

Estimated Authors of the Case Report "Chronic Inflammatory Demyelinating Polyneuropathy after ChAdOx1 nCoV-19 Vaccination",

I've read your paper with great interest. Since the inception of the global vaccination campaign against SARS-CoV-2, increasing claims on neurological side effects of vaccination (possibly related to some characteristics of the spike protein, whom the vaccine-induced antigens were modelled from) have been reported.

Therefore, your case report may be of certain interest for clinician working in the vaccination settings.

Unfortunately, in my opinion at least, your paper fails in fulfilling the aims of a case report for the following reasons:

1) the potential association between SARS-CoV-2 vaccine (and mostly adenovirus-based formulates) and GBS has been previously described (see https://www.ncbi.nlm.nih.gov/pmc/articles/PMC8196284/; https://pubmed.ncbi.nlm.nih.gov/34750810/) and some high quality review have been published on this topic. Therefore, the present subject is not totally new.

2) your case report, despite its inherent and undisputed interest, fails in explaining why it may deserve readers' attention. More precisely: while the case is substantially well described, discussion should explain to the reader what makes this case new and different from other GBS cases; 

3) please include the "table" that is cited across the text and not included in the main file.

Author Response

I've read your paper with great interest. Since the inception of the global vaccination campaign against SARS-CoV-2, increasing claims on neurological side effects of vaccination (possibly related to some characteristics of the spike protein, whom the vaccine-induced antigens were modelled from) have been reported.

Therefore, your case report may be of certain interest for clinician working in the vaccination settings.

Response: Thank you for your comment.

Unfortunately, in my opinion at least, your paper fails in fulfilling the aims of a case report for the following reasons:

1) the potential association between SARS-CoV-2 vaccine (and mostly adenovirus-based formulates) and GBS has been previously described (see https://www.ncbi.nlm.nih.gov/pmc/articles/PMC8196284/; https://pubmed.ncbi.nlm.nih.gov/34750810/) and some high quality review have been published on this topic. Therefore, the present subject is not totally new.

Response: We agree with the reviewer that several cases of GBS following COVID-19 vaccination have already been described and therefore our report appears to confirm previous studies. However in this case a prolonged clinical follow up allowed to diagnose a chronic form of demyelinating polyneuropathy, that is an unexpected clinical evolution. We think that this finding is new and has not been described in the available reviews of neurological adverse events following SARS-CoV-2 vaccination.

2) your case report, despite its inherent and undisputed interest, fails in explaining why it may deserve readers' attention. More precisely: while the case is substantially well described, discussion should explain to the reader what makes this case new and different from other GBS cases;

Response: Thank you for this remark. We changed the discussion to underline the originality of the case report and the clinical interest in the field of SARS-CoV-2 vaccines.

3) please include the "table" that is cited across the text and not included in the main file.

Response: Thank you for your remark. We added the table to the new version.

Reviewer 2 Report

The authors present a case of a patient with chronic inflammatory demyelinating polyneuropathy after ChAdOX1 nCOV-19 vaccination. The case report contains information that is in the very focuse of the current scientific interest within the COVID-19 field. The presentation of the case is detailes and the results of the brain and spinal cord MRI findings are avalable as well. Diagnostic and treatment approach are outlined in detailes. This report clearly shows the need for close monitoring of patients developing Guillain Barre syndrome after COVID-19 vaccination after the acute phase and represents an important contribution to the field, particularly in the specific field of vector-based SARS-CoV-2 vaccines. 

I thing the case report should be published but following major revision: 

Main concerns:

  1. Section Case presentation. Please provide information on the the scope and methods used to exclude CNS infections. When describing a patients with diagnosis of Guillain Barre syndrome, particularly in the context of COVID-19 vaccination, it is essential to include detailed information on microbiological work-up of the CSF. A negative results of a molecular multiplex CNS panel (many IVD assays of thi type are available on different molecular platforms) is needed to confirm the lack of infection. The absence of pleocytosis and results of biochemical assays are strong arguments for the lack of infection at the time of analysis, but evidence-based medicine (particularly in the context of the significance of the data presented in the case) realy needs to be confirmed with objective testing. Therefore, please provide results of microbiological assays documenting the lack of infection.
  2. In section Clinical presentation, line 82, the authors state "Clinical features and examinations of the patient are summarised in table." However, in the PDF that I recieved for review, there is not table (just Figure 1). Therefore, please solve this issue.
  3. It would be very usefull to include a table with selected characteristics of patients described in references 8-12 (with GBS follosing CoV-19 vaccination) the authors mention in section Introduction.  
  4. In section Introduction, lines 27-29, the information "more than four million doses administrated worldwide" needs to be updated. The current number of administrated doses worldwide is in fact 7,548,676,211 (as of November 17th 2021), based on John Hopkins University COVID-19 dashboard. Please update the information and provide a correct reference for the statement.

Author Response

The authors present a case of a patient with chronic inflammatory demyelinating polyneuropathy after ChAdOX1 nCOV-19 vaccination. The case report contains information that is in the very focuse of the current scientific interest within the COVID-19 field. The presentation of the case is detailes and the results of the brain and spinal cord MRI findings are avalable as well. Diagnostic and treatment approach are outlined in detailes. This report clearly shows the need for close monitoring of patients developing Guillain Barre syndrome after COVID-19 vaccination after the acute phase and represents an important contribution to the field, particularly in the specific field of vector-based SARS-CoV-2 vaccines. 

Response: Thank you for your comments and opinions.

I think the case report should be published but following major revision: 

Main concerns:

Section Case presentation. Please provide information on the the scope and methods used to exclude CNS infections. When describing a patients with diagnosis of Guillain Barre syndrome, particularly in the context of COVID-19 vaccination, it is essential to include detailed information on microbiological work-up of the CSF. A negative results of a molecular multiplex CNS panel (many IVD assays of thi type are available on different molecular platforms) is needed to confirm the lack of infection. The absence of pleocytosis and results of biochemical assays are strong arguments for the lack of infection at the time of analysis, but evidence-based medicine (particularly in the context of the significance of the data presented in the case) realy needs to be confirmed with objective testing. Therefore, please provide results of microbiological assays documenting the lack of infection.

Response: Thank you for your suggestion. We agree with the reviewer and we added the lacking informations in the case presentation.

In section Clinical presentation, line 82, the authors state "Clinical features and examinations of the patient are summarised in table." However, in the PDF that I recieved for review, there is not table (just Figure 1). Therefore, please solve this issue.

Response: Thank you for your remark. We added the table to the new version.

It would be very usefull to include a table with selected characteristics of patients described in references 8-12 (with GBS follosing CoV-19 vaccination) the authors mention in section Introduction.

Response: Thank you for your remark. We added to the table the characteristics of the patients described in the studies 8-12.

In section Introduction, lines 27-29, the information "more than four million doses administrated worldwide" needs to be updated. The current number of administrated doses worldwide is in fact 7,548,676,211 (as of November 17th 2021), based on John Hopkins University COVID-19 dashboard. Please update the information and provide a correct reference for the statement.

Response: Thank you for your remark. We changed the sentence and the reference as requested.

Round 2

Reviewer 2 Report

The authors responded to all suggestions and revised the manuscript accordingly. The added table has provided exceptionally important information that will allow the readers to put the data presented in the case report into an appropriate context.

The manuscript is now ready for publication in the present form.